# Quality of Life of Schoolchildren Living with a Long-Term Sick Parent: The Role of Tasks at Home, Life Circumstances and Social Support

**DOI:** 10.3390/ijerph19127043

**Published:** 2022-06-08

**Authors:** Simone A. de Roos, Jurjen Iedema, Alice H. de Boer

**Affiliations:** 1Programme Participation, Talent Development, and Equality of Opportunity, SCP, The Netherlands Institute for Social Research, Bezuidenhoutseweg 30, 2594 AV The Hague, The Netherlands; j.iedema@scp.nl (J.I.); a.de.boer@scp.nl (A.H.d.B.); 2Faculty of Social Sciences, Vrije Universiteit Amsterdam, 1081 HV Amsterdam, The Netherlands

**Keywords:** schoolchildren, long-term sick parent, life satisfaction, psychosomatic complaints, quality of life, intensity of tasks, social support, young carers

## Abstract

This study investigates whether there are differences in quality of life—i.e., psychosomatic complaints and life satisfaction—between schoolchildren with and without a chronically ill or disabled parent at home. It also examines the role played by the intensity of tasks, life circumstances, and social support at home and school. In 2017, a Dutch representative sample of adolescents aged between 12 and 16 (from the Health Behaviour in School-aged Children study) completed a questionnaire about illness of family members, tasks at home, life circumstances and characteristics, social support, psychosomatic complaints and life satisfaction. In total, 5470 schoolchildren who did not have a parent with a chronic illness and 652 who did have a parent with a chronic illness were selected (average age 13.9). Stepwise multilevel logistic regression analyses in STATA were used. Schoolchildren with an ill parent had more psychosomatic complaints and lower life satisfaction than their counterparts without an ill parent, even when controlling for extra task hours, specific life circumstances and characteristics (e.g., more likely to be growing up in a single-parent family or stepfamily and more likely to be female), and lower perceived support. These aspects are also predictors of a lower quality of life. Professionals should address these aspects of the life of schoolchildren with a sick parent in such a way that they are facilitated to make a successful transition to adulthood.

## 1. Introduction

A significant proportion of schoolchildren in the Netherlands live with a chronically ill family member (19%) [1]. International research shows that the number of young people dealing with illness and care situations in the family is increasing, as is the case in Great Britain [2]. Previous studies revealed that young people who are confronted with illness in their family have less favourable developmental outcomes compared to peers who do not have an ill family member: they experience more stress and emotional issues, have more problems at school and skip classes more often [1,3,4,5,6,7,8,9]. Although many young people grow up with ill brothers or sisters and live-in grandparents or other family members, over half of the cases with a care situation involve an ill parent [1]. There are indications that the social relationship is important for quality of life: living with a chronically ill parent is more burdensome on young people’s well-being than a brother or sister or live-in grandparent with a long-term illness [9,10,11]. This publication therefore focuses on schoolchildren with a chronically ill parent. Based on a large representative sample of schoolchildren, we compare the quality of life of young people with a chronically ill parent to the quality of life of young people who do not have an ill parent at home (or another ill family member). With regard to quality of life, we focus on psychosomatic complaints as well as life satisfaction. Health complaints are indicative of subjective health. Life satisfaction reflects the subjective evaluation of quality of life over all life domains. Prior research has shown that few psychosomatic complaints and a high degree of perceived life satisfaction point to a higher quality of life [12]. The central question is: Are there differences in quality of life—life satisfaction and psychosomatic complaints—between young people with and without a chronically ill parent at home and, if so, what are these differences and how can they be explained? The young people in this study are schoolchildren aged 12 to 16 who attend secondary school. A chronically ill parent refers to a parent in the family with whom the young person spends most of their time who has been physically and/or mentally ill or disabled for an extended period (at least three months).

An increasing amount of research is being published about young people with an ill parent at home, including their helping behaviour, the negative effects that this situation can have on the quality of life of those involved and what policies and supporting measures can do to alleviate such effects. Nevertheless, the desire is also frequently expressed to conduct more theory-driven research, as this increases the generalisability of results [13,14]. This study helps to build theory-driven knowledge by examining the differences in quality of life via hypotheses regarding: the care situation (1), intensity of tasks (2), living conditions and characteristics (3) and support (4).

### 1.1. Care Situation

First, we investigate whether young people with a chronically ill parent report a lower quality of life than young people without an ill parent and, if so, whether this can be explained by the effect that growing up with illness in itself may have on schoolchildren’s quality of life, for instance because they worry about the ill parent [14]. This reasoning as well as the other hypotheses are deduced from the ecological model of Bronfenbrenner [15]; a broad developmental theory describing the development of a child as a dynamic process between the child, parent(s), other significant persons (such as friends and teachers) and the (broader) environment including living conditions. According to this model, a care situation at home is an unfavourable condition that creates a risk for lower quality of life of young people [15,16,17]. There are indications that young people with an ill parent experience a great deal of stress, worries, fears, anger and sadness about the ill parent’s limitations, needs and pain, as a result of which the schoolchildren themselves experience a lower quality of life [6,7,9]. Studies of children whose parents have psychological problems or an addiction also show that they themselves are at an increased risk of developing a psychological disorder or falling into addiction at some point [18]. In addition to worrying about the ill parent, this could also be related to genetic effects. De Roos and colleagues [19] showed that the well-being of students with an ill family member (including ill parents) is less favourable than that of their peers, correcting for socio-demographic characteristics. These authors used a narrow definition of low well-being, in terms of having more psychological problems, such as emotional and behavioural problems.

In this study, we examine whether living with an ill parent is related to schoolchildren’s quality of life, independent of the intensity of tasks at home, young people’s living conditions and characteristics and available support. This brings us to our first hypothesis, which relates to the *care situation* (also known as the ‘family effect’) [14]:

**Hypothesis** **1** **(H1).***Schoolchildren dealing with a chronically ill parent experience a lower quality of life than schoolchildren without an ill parent, even if we take into account the intensity of tasks at home, living conditions and characteristics and support from the surrounding environment*.

### 1.2. Intensity of Tasks at Home: Extra Task Hours

Second, we examine the extent to which a possible difference in quality of life between young people with and without an ill parent can be explained by the performance of extra task hours, i.e., ‘caring’ [20]. Bobinac and colleagues [14] called this the *informal care effect*. Fisher and Tronto [21] specified caring as ‘taking care’ (taking initiative and responsibility for care tasks) and ‘caregiving’ (actually performing care tasks). The effects of caring—perceived stress, burden and health—have been extensively studied in adults providing informal care, but rarely in young people [7,22]. Theoretically, informal care by young people is seen as a risk factor for low quality of life from the perspective of parentification (role reversal) [23]. From a life-course theoretical perspective (based on the ecological model [16]), the idea is, therefore, that young people take on responsibilities that are not appropriate for their age and identity development [24,25]. The tasks can also take up so much time and/or energy that young people hardly have time for themselves and for relaxation, which can lead to a low quality of life [19,22]. Previous studies on tasks and responsibilities at home have mainly examined young people growing up with an ill family member or young people who already provided informal care, often without using a comparison group. Such research tends to focus on the frequency, intensity or nature of informal care tasks [13,19,26]. However, very few studies have examined whether the tasks performed by these young people at home differ from those of young people without illness and care obligations in the family. There are exceptions, such as a study which showed that young people with a physically ill parent were more likely to perform care-related and household tasks than young people without an ill family member [7]. Because this study involved a small and non-representative sample and did not address the intensity of the help, the results have limited generalisability.

In the present study, we look at the performance of (extra) task hours among all young people and whether this is associated with (a difference in) quality of life [7,19,22]. Although we are only able to examine the relationship between informal care and quality of life in young people with an ill parent, other young people could take on additional (adverse) tasks and responsibilities as well. It is known, for instance, that young people with a migration background are relatively likely to have responsibilities within the family [27]. Research by de Roos and colleagues [19] on the relationship between informal care and well-being found that schoolchildren with an ill family member (including those with an ill parent) develop more psychological problems as they provide more household and administrative help or intensive help. Based on these findings, we can formulate a second hypothesis about the role of extra tasks, also known as the *informal care effect* [14]:

**Hypothesis** **2** **(H2).**
*Schoolchildren dealing with a chronically ill parent experience a lower quality of life because they spend more time on tasks at home than schoolchildren who do not have an ill parent (even after correcting for living conditions and characteristics and support from the surrounding environment).*


### 1.3. Living Conditions and Characteristics

Regarding the third explanation for possible differences in quality of life related to whether or not schoolchildren are living with an ill parent, we will look at young people’s living conditions and characteristics. According to the ecological model, these are also important factors in young people’s quality of life [16]. There are indications that young people who grow up in a family with an ill parent often live in unfavourable conditions [11]. For example, prior research has shown that among 13- to 23-year-olds with ill family members (including ill parents), low socio-economic groups (low education and low family wealth, migration background and growing up in single-parent and step-families) and girls are overrepresented [22,28,29]. In a qualitative focus group study with young informal carers in Canada, Stamatopoulos [30] described the so-called ‘young care penalty’: ‘those from single parent and single child families, dealing with more stigmatized and debilitating problems had the highest penalty’. We also know from prior research that these socio-demographic factors (in addition to older age) increase the risk of a lower quality of life among young people [31,32]. Recent studies suggest that lower quality of life among girls can be partly explained by the fact that girls provide more help [33].

In addition to gender, age, migration background, education level, family wealth and family structure, we also consider the importance that young people attach to religion. This is in light of findings from previous studies that the degree of religiosity seems to make a difference in quality of life: young people who do not consider religion to be important experience a lower quality of life (including in terms of having more psychological problems) than young people who do care about religion [19,34,35]. There are indications that religious values and norms are related to the provision of informal care in general, as these are seen as a motivating factor to provide help and continue doing so [36]. Dearden and Becker [37] showed that some young Muslims are against ‘cross-gender care’, which can have major consequences if the young person is the only suitable helper and must care for a loved one of the opposite gender. We view these research findings as a rationale for including religiosity in the present study. Therefore, we aim to include a wide variety of living conditions and characteristics. Our third hypothesis is as follows:

**Hypothesis** **3** **(H3).**
*Schoolchildren dealing with a chronically ill parent experience a lower quality of life because they are more likely to grow up in unfavourable living conditions than young people without an ill parent, or because they have characteristics that increase their risk of a lower quality of life (even after correcting for extra task hours and support from the surrounding environment).*


### 1.4. Support

Fourth, according to the transactional development model and life course model (both are based on the ecological model [16]) and social resource theory, among others, support from the social environment is crucial to young people’s quality of life, as there is also abundant empirical evidence of this [17,25,38,39,40,41,42,43]. In the transactional development model and the life course model the focus lies on completing successive development tasks (e.g., developing one’s own identity and obtaining a basic qualification) and the role of support from parents and significant others in this regard. The social resource model gives information on how the structures of provision effects the positive or negative outcomes of informal care giving. The present study focuses on support within the family from parents, brothers and sisters, as well as support from friends, classmates and teachers [32,44]. Previous quantitative studies on the relationship between growing up with a care situation and well-being have devoted some attention to support at home and support from peers. These studies focused on problems experienced at home [11], mutual family ties [18], openness of communication with parents [45] and connection with peers [29]. In these studies, young people with a care situation at home (not only with ill parents) scored worse than their peers without a care situation. In line with expectations based on the transactional developmental model and life course model, in a stressful situation such as illness in a family, parenting and the parent–child relationship can come under pressure. Parents then may not have enough time and space to give their child optimal attention. Additionally, young people with a sick parent may not have enough time and space to meet up with friends, and as a result, may find it more difficult to connect with peers than average young people [29]. Little is known about perceived support from teachers in a care situation. We do know, however, that young informal carers would like their teachers to notice their situation sooner and be more understanding of it [46]. In this publication, we assume that schoolchildren with an ill parent experience less support than their peers, as prior research has shown that young people with a care situation at home (including ill parents) more often feel alone [11].

There has long been discussion in academic forums about the mechanisms underlying the role of *support* [25]. On the one hand, there seem to be direct effects: social support directly improves quality of life, for example because this strengthens young people’s self-esteem, confidence and resilience. At the same time, there may be a buffer effect as well: support protects against or reduces the potential harmful effects of growing up with care obligations [47]. Our fourth hypothesis is formulated as follows:

**Hypothesis** **4** **(H4).***Schoolchildren who grow up with an ill parent at home experience a lower quality of life because they experience less support from their environment than schoolchildren without an ill parent, even after correcting for extra task hours at home and living conditions and characteristics*.

## 2. Materials and Methods

### 2.1. Participants

Quantitative data on Dutch young people from the Health Behaviour in School-aged Children study (HBSC, data from HBSC’17) [32,48] were used. This is a representative international survey that is conducted every four years among school-aged youth from 11 to 16 years old. The survey was administered to over 8000 schoolchildren at 72 primary schools and 85 secondary schools in the Netherlands [32]. For this article, only the data on secondary school students have been included. As for growing up while dealing with illness, only those young people with a chronically ill parent have been selected. The response rate of secondary schools was 37%. The data pertain to 6122 young people, 652 of whom have a chronically ill parent. The average age is 13.9 years. See Table 1 for more characteristics of the sample.

### 2.2. Procedure

The sample of schools was randomly taken from a file listing all regular primary and secondary schools in the Netherlands, compiled by the Education Executive Agency of the Ministry of Education, Culture and Science. The distribution of schools in urban and rural areas was taken into account during the sampling process. Within the participating schools, one class from each year was then randomly selected for participation.

The questionnaires were administered in a classroom-based test setting under the supervision of research assistants in the fall of 2017. The schoolchildren completed the survey digitally and anonymously. Informed consent was obtained from schools, adolescents and their parents and ethical approval was given by the Ethics Assessment Committee of the Faculty of Social Sciences at Utrecht University (FETC17-079 in 2017) for the Dutch HBSC study.

### 2.3. Measures

#### 2.3.1. Chronically Ill Parent at the Schoolchild’s Home

The question about having a chronically ill family member was formulated in the HSBC survey as follows: ‘Is there someone in your home (the house or family where you spend most of your time) who has been physically and/or mentally ill or disabled for longer than three months? Examples of illnesses include: cancer, diabetes, heart disease, depression, addiction, autism, intellectual disability.’ The answer options were: yes, myself; yes, my father/mother; yes, my brother/sister; yes, a grandparent who lives with us; yes, someone else; no, no one. Young people who indicated having an ill father or mother and young people without an ill parent at home were included in the analysis. In the analyses, no distinction is made between having an ill mother or an ill father. The schoolchildren dealing with an ill sibling or other ill family member (not a parent) were not included in the analyses.

#### 2.3.2. Tasks at Home

In the HSBC survey, all schoolchildren were asked about performing various tasks: ‘Do you have any tasks or responsibilities at home (the house or family where you spend most of your time), such as cleaning the house, helping a younger brother or sister with homework or personally taking care of an ill family member?’ If the answer was affirmative, the young people were asked a follow-up question about the intensity of the tasks: ‘Approximately how many hours per week have you spent on tasks at home in the past month?’ The answer options were: less than 1 h (scored as 0.5 h), 1 h, 2 or 3 h (2.5), 4 or 5 h (4.5), 6 or 7 h (6.5) and 8 h or more (8). Young people who indicated that they did not perform any tasks were given a score of 0 h. ‘Extra’ task hours is defined as performing more tasks than usual, where spending an average of more than four hours performing tasks in the month prior to the survey is taken as the cut-off point. This threshold value is based on the total group and lies one standard deviation above the mean. A similar intensity threshold was also used in the study by Ruijter [49] on students providing informal care.

#### 2.3.3. Living Conditions and Characteristics

Seven life circumstances and characteristics were measured: age, gender, school level, ethnicity, family affluence, household situation and importance of religion. Respondents entered their date of birth, gender (‘Are you a boy or a girl?’, no other option was provided in the questionnaire) and the level of education they were currently following (ranging from 1 = pre-vocational secondary education (vocational pathway) (low) to 4 = pre-university education (high)) and in which year they were. They also stated in which country they and their parents were born. In line with the Dutch standard for coding ethnicity, adolescents who are born in another country or who have a parent who was born in another country were categorised as having a migrant background. The affluence of the respondent’s household was measured using the Family Affluence Scale (FAS-III) [32,50]. This includes questions about material possessions (the number of cars and computers), characteristics of the house (whether they have their own bedroom, the number of bathrooms and whether it has a dishwasher) and the number of times the family has been on holiday in the past year. Based on the sum score for these questions (0–13), three groups were distinguished: adolescents with low (0–6), average (7–9) or high family affluence (10–13) [32,51]. With regard to their household situation, respondents indicated whether or not they were living in a complete family (both biological or adoptive parents present in one house). They also stated whether they were brought up in a certain faith and how important religion is for them. Two groups were distinguished: no religious upbringing/religion is not (very) important versus religion is (somewhat or very) important.

#### 2.3.4. Social Support

The survey asked about perceived support in schoolchildren’s private life (from those at home and from friends, based on the shortened version of the Multidimensional Scale of Perceived Social Support (MSPSS) [52] and school setting (from teachers and classmates) [53].

To this end, the young people were presented with four statements about the perceived *support at home*, such as ‘the people in my family really do their best to help me’ and ‘I can talk about problems at home’. The young people indicated to what extent they agreed with the statements based on a seven-point scale (1 = completely disagree to 7 = completely agree). As these variables were highly skewed (all skewness measures were < −1.8), we used confirmatory factor analysis treating the variables as ordinal. The fit measures of the one factor solution were satisfactory (Root Mean Square Error of Approximation (RMSEA) = 0.10, Comparative Fit Index (CFI) = 1.00, Tucker–Lewis index (TLI) = 1.00). To use the scale in further analyses, factor scores were computed and these were transformed linearly into values between 1 and 7 inclusive, to enable a similar interpretation as the original variables. Higher scores indicate higher levels of support at home.

The perceived *support from friends* was also assessed based on responses to four statements, once again using a seven-point scale. A sample item: ‘my friends really try to help me’. The answers to these statements were also highly skewed (all skewness measures were <−1.2), and therefore, a confirmatory factor analysis was used once again. The fit measures of the scale were satisfactory, except for the RMSEA (RMSEA = 0.22, CFI = 1.00, TLI = 0.99). The RMSEA was too high, but the analysis could not be improved (all four standardised factor loadings were higher than 0.91). To use the scale in further analyses, once again, factor scores were computed and these were transformed linearly into values between 1 and 7 inclusive. Higher scores indicate higher levels of support from friends.

To assess the *support from teachers*, young people were presented with three statements, such as ‘I feel that my teachers accept me the way I am’. Young people responded using a five-point scale to indicate how much they agreed with the statements (1 = completely disagree to 5 = completely agree). The answers on two of the statements were moderately skewed (about −0.55) and were highly skewed on one statement (<−1.2). Therefore, another confirmatory factor analysis was used. The fit measures of the scale were satisfactory (RMSEA = 0.00, CFI = 1.00, TLI = 1.00). To use the scale in further analyses, factor scores were computed and these were transformed linearly into values between 1 and 5 inclusive. Higher scores indicate higher levels of support from teachers.

To measure the perceived *support from classmates*, young people were also presented with three statements, such as ‘Other classmates accept me the way I am’. The schoolchildren indicated their agreement with each statement using a five-point scale. As the answers to these statements were moderately to highly skewed (one measure of −0.87 and two of <−1.2, respectively), a confirmatory factor analysis was used once again. The fit measures of the scale were satisfactory (RMSEA = 0.00, CFI = 1.00, TLI = 1.00). To use the scale in further analyses, again, factor scores were computed and these were transformed linearly into values between 1 and 5 inclusive. Higher scores indicate higher levels of support from classmates.

#### 2.3.5. Psychosomatic Complaints

Respondents were asked if they have suffered from ten psychosomatic complaints in the last six months (e.g., headache, stomach ache, feeling low, sleeping difficulties, feeling exhausted). They responded on a five-point scale (ranging from 1 = rarely or never to 5 = about every day) [54]. The answers on five items were moderately skewed (between −1.0 and −0.5) and were highly skewed (<−1.0) on the other five items. Therefore, a confirmatory factor analysis was used. The fit measures of the scale were satisfactory (RMSEA = 0.11, CFI = 0.94, TLI = 0.92). To use the scale in further (bivariate and multivariate) analyses, factor scores were computed and these were transformed linearly into values between 1 and 10 inclusive. Higher scores indicate higher levels of psychosomatic complaints. For the bivariate analyses (in Section 3.1), scores on the separate complaints were also rescaled into values between 1 and 10 inclusive.

#### 2.3.6. Life Satisfaction

Cantril’s self-anchoring ladder was used to measure life satisfaction, a well-known measure and repeatedly used in the HBSC survey [48,53,55]. Respondents were asked to rate their current life satisfaction using a ladder with steps numbered from 0 (extremely unsatisfied) to 10 (extremely satisfied).

### 2.4. Statistical Analysis

Bivariate and multivariate analyses were performed. First, *F*-tests were used to measure differences between adolescents without and with a sick parent on performing extra task hours at home, life circumstances and characteristics, social support and quality of life. Additionally, the correlation between the two aspects of quality of life was computed, yielding a negative correlation between psychosomatic complaints (based on the scale) and life satisfaction (*r* = −0.46, *p* < 0.000). Because of the shared variance of these aspects is relatively low (*r*^2^ × 100 < 25%), these outcome measures were considered separately in the multivariate analysis.

To find out what role having an ill parent plays in psychosomatic complaints (based on the scale) and life satisfaction (in addition to having or not having extra task hours at home, living conditions and characteristics and perceived support), we then performed stepwise linear multilevel regression analyses. We used multilevel analyses because the HBSC survey respondents are nested in schools and classes. The analyses, therefore, control for school and class level. First, we included having an ill parent as an explanatory factor of psychosomatic complaints and life satisfaction, and then, we added extra task hours. This provides insight into the relative importance of the care situation at home for the outcome measure compared to extra task hours at home. Next, we examined a possible substitutive or additive effect of living conditions on the outcome measure. In the last model, we added support characteristics as an explanatory factor.

Stata version 16 was used for the analyses. As there was very little explained variance at the school level, we used a two-level model with a class and an individual level, cluster-correcting the standard errors for the school level, with the added advantage that we could weight the data (enabling correct statements about the Dutch adolescent population). Weights were based on gender, grade, educational type and urbanisation.

## 3. Results

### 3.1. Descriptives

The details of the subgroups adolescents without a sick parent and adolescents with a sick parent can be found in Table 1.

The groups differed on extra hours of tasks performed at home. Schoolchildren with a sick parent at home were more likely to spend extra hours on tasks at home (about 19%) than their counterparts without a sick parent (about 12%). Additionally, background differences were found for gender, family affluence, and not living with both parents in one house. Adolescents with a sick parent were more likely to be female (57% vs. 47%), less likely to have grown up in families having high affluence (35% vs. 43%) and more likely not to be living with both parents in one house (26% vs. 22%) than their peers without an ill parent. The groups also differed on all support measures except for support from friends. The schoolchildren having a sick parent experienced less support at home, less support from teachers and less support from classmates than those having no ill parent at home. Finally, they had a higher score on psychosomatic complaints (scale) and a lower score on life satisfaction. The groups also differed on all separate psychosomatic complaints (such as feeling low and feeling tired), showing higher scores among the schoolchildren with a sick parent at home.

### 3.2. Psychosomatic Complaints

All linear multilevel regression analyses yielded a significant association between having a sick parent and psychosomatic complaints (based on the scale) (Table 2). In Model 1, living with a parent with a long-term illness explains around 1% of the differences in psychosomatic complaints (*b* = 0.58), in addition to 4% explained by the class level (ICC = 4%). The care situation is slightly less correlated if the extra task hours are included in the analysis (*b* = 0.55, Model 2), but it is still significant. This feature is associated with a higher likelihood of psychosomatic complaints, and adds half a percentage point of explained variance (for a total of just under 2%). These findings support the first hypothesis regarding the role of the care situation, and also partly support the second hypothesis regarding the role of informal care: young people with an ill parent have slightly more psychosomatic complaints because they are slightly more likely to spend extra hours on tasks at home than young people without an ill parent (see also Table 1).

When living conditions and characteristics are added in Model 3, the correlation between having an ill parent and experiencing psychosomatic complaints once again becomes slightly weaker, but remains significant. The link between extra task hours and psychosomatic complaints likewise diminishes somewhat, but remains significant. The first two hypotheses are therefore partly supported. The total explained variance is now 7.5%: a significant addition of just under six percentage points compared to Model 2. In addition to having an ill parent and performing extra task hours, the main factors associated with psychosomatic complaints are gender, age, ethnicity, family structure and the importance that young people attach to religion.

Girls (*b* = 0.77) and older students (*b* = 0.10), for instance, are more likely to report such complaints than boys and younger schoolchildren. Young people with a migration background and young people who do not live in the same house with both parents are more likely to suffer from psychosomatic complaints than young people who do not have a migration background and who do live with both parents. Furthermore, young people who believe religion is important are less likely to report suffering from psychosomatic complaints than young people who consider religion unimportant (*b* = −0.19). These results are partly consistent with Hypothesis 3: young people with an ill parent are more likely to have characteristics or to grow up in living conditions that increase the risk of psychosomatic complaints than young people without an ill parent, in this case being a girl and not living in the same house with both parents (e.g., in a single-parent household or with a step-family).

After including support from the four types of sources in the analysis of Model 4, the correlation between having an ill parent and psychosomatic complaints once again decreases, but remains significant. This is yet another confirmation of Hypothesis 1: the care situation plays a small, but significant role. The correlation between extra task hours and psychosomatic complaints likewise decreases, but remains significant as well. The relationship between living conditions and characteristics on the one hand and psychosomatic complains on the other becomes weaker in general, but remains significant after adding the support variable to the analysis. Support appears to be an important protective factor against psychosomatic complaints, since schoolchildren’s risk of psychosomatic complaints decreases as they receive more support from various sources. In total, the full Model 4 explains approximately 17% of the differences in psychosomatic complaints among schoolchildren, which includes more than nine additional percentage points after adding support. This makes support the most robust predictor of psychosomatic complaints. These results also partly confirm Hypothesis 4 regarding the role of support: young people with an ill parent experience somewhat less support at home, from teachers and from classmates than young people without an ill parent (Table 1), which is partly why they also experience more psychosomatic complaints. Furthermore, the class which the young person is in still explains 2% of the differences in psychosomatic complaints.

### 3.3. Life Satisfaction

Table 3 shows the results of the multivariate linear multilevel regression analyses, which include extra task hours (Model 2), living conditions and characteristics (Model 3) and support (Model 4) as predictors of life satisfaction, in addition to having an ill parent.

The care situation at home in Model 1 is found to be a unique predictor of life satisfaction, even after including the other variables (Models 2–4), which again (as in Section 3.2) points to the role of the care situation (Hypothesis 1). The care situation explains just under 1% of the variance in life satisfaction (Model 1) and the classes which young people are in explains 5%. When extra task hours are added, the correlation between the care situation and life satisfaction becomes slightly weaker, but remains significant (thus continuing to support Hypothesis 1). Model 2 explains around 1% of the variance, with a very small yet significant addition of one-third of a percentage point due to the inclusion of extra task hours. Young people who spend additional time on tasks at home are less satisfied with their lives than young people who do not do this. The results suggest a small role played by informal care (Hypothesis 2): because the intensity of tasks is higher among young people with an ill parent (see Table 1), they score slightly lower on life satisfaction than their peers without an ill parent.

In Model 3, we also include living conditions and characteristics in the analysis, and while these factors decrease the coefficients of the care situation and extra task hours, both remain significant (thus again confirming Hypotheses 1 and 2). They increase the explained variance by over five percentage points (with more than 6% of life satisfaction now explained). As with psychosomatic complaints, gender is once again a key predictor of life satisfaction: girls rate their lives half a point lower on average than boys (7.4 versus 7.9, not presented) (*b* = −0.50). Age is also related to life satisfaction: the results show that young people become less satisfied with their lives as they grow older (see also Inchley and colleagues [48]). It is interesting to note that in Models 3 and 4, origin is not related to life satisfaction (after controlling for a series of characteristics); this relationship was present, however, in the analysis of psychosomatic complaints (Table 2). Furthermore, schoolchildren who grow up with low or average family wealth are less satisfied with their lives than schoolchildren from families with a high level of wealth (*b* = −0.45 and *b* = −0.23, respectively). Schoolchildren who do not grow up in the same house with both parents and who attend an average to high level of schooling are also less satisfied with their lives than their peers who do grow up with both parents and who attend the highest level of schooling. In addition, as with psychosomatic complaints, religion seems to make a difference: students who believe religion is important are more positive about their lives than those who believe otherwise (*b* = 0.22). These results partly indicate the effect of unfavourable living conditions and characteristics (Hypothesis 3): because young people with an ill parent are more likely to be female, less likely to live in the same house with both parents and less likely to come from families with a high level of wealth (see Table 1), these young people are less positive about their lives than young people without an ill parent.

Adding support in Model 4 increases the degree to which life satisfaction is explained, by 17 percentage points. This makes support the most robust predictor of life satisfaction, as is also the case with psychosomatic complaints. In total, nearly 25% of the variance is now explained. There is a positive correlation between all the different sources of support on the one hand and life satisfaction on the other. Support at home seems to have the most influence on the score that schoolchildren assign to their lives (this is where the regression coefficient is highest, *b =* 0.30). Interestingly, we also see a strong flattening of the regression coefficient for having an ill parent, after adding support variables. This is an indication of an indirect effect: part of the relationship between an ill parent at home and life satisfaction likely depends on the degree of perceived support (Hypothesis 4). Because young people with an ill parent experience less support at home and at school, they are somewhat less positive about their lives. However, the effect of having an ill parent remains significant, which is an indication of Hypothesis 1. Additionally, a small portion of the differences in the degree of life satisfaction is explained by the classes which schoolchildren are in (this explains 1%).

## 4. Discussion

The purpose of the present study was to assess the quality of life of schoolchildren who are living with a chronically ill parent, by comparing them with their peers who are not in this situation and testing specific hypotheses to explain differences between these groups. Our results showed that adolescents who had a chronically ill parent had lower quality of life—i.e., had more psychosomatic complaints like fatigue and sleeping problems—and had lower life satisfaction than their peers without a chronically ill parent. The relationship between this care situation and both aspects of quality of life remains significant after taking into account extra task hours, life circumstances and characteristics and perceived support—but it is small. This implies that the presence of a sick parent at home—including the likely worries about the sick parent and the uncertainty this brings regarding the future—partly contributes to their increased psychosomatic complaints and lower life satisfaction (H1). These findings indicate the presence of a small role of the care situation for both aspects of schoolchildren’s quality of life [9,14].

There are also some indications that task hours, described as informal care, also play a role [7,14] (H2). Compared with their peers without a sick parent, schoolchildren with a sick parent at home devote extra time to tasks, as a result of which they report a slightly lower quality of life. However, although the role that informal care plays is significant, it is not very big, which appears to suggest that this effect is weaker than the effect of life circumstances and characteristics (H3) or the role that support (H4) plays.

One of the characteristics that has the strongest relation to quality of life is gender. Girls are particularly at risk of low life satisfaction and high levels of psychosomatic complaints, both in the population as a whole as well as among those with a sick parent [19,32]. One reason for this could be the complex interplay of biological, psychological and social changes during puberty. For example, culturally determined and gendered role expectations and socialisation in relation to dealing with emotions (e.g., more internalisation among girls, more externalisation among boys) may lead girls to experience more psychosomatic complaints and lower life satisfaction [56,57]. Girls are relatively more likely to report a chronically ill parent at home and since they more often report lower life satisfaction (see Table 1); gender partly explains the relationship between having a sick parent and lower life satisfaction. The finding that girls are more likely to indicate that they have a sick parent than boys is in line with findings from a systematic review based on qualitative and quantitative research [4]. It is possible that girls are more sensitive to and more rapidly pick up signals of a need for care on the part of a parent or other relatives or loved ones, or it may be that boys are more reluctant to report such a family situation in a survey or interview [57]. It is also possible that more is expected of girls as regards keeping an eye on the wellbeing of relatives. This would suggest gender-specific role expectations which are transferred during the socialisation process [56,58].

Schoolchildren who do not live with both parents in one house (e.g., from single-parent families and stepfamilies) as well as those growing up in less affluent households also report lower life satisfaction—which has also been repeatedly found in other studies [31,32]. Adolescents with a chronically ill parent at home are overrepresented in these groups, which also partly explains their lower quality of life (H3). Schoolchildren’s quality of life is not just related to the characteristics of gender, family affluence and family structure. It is, for example, striking that the importance schoolchildren attach to religion acts as a protective factor against psychosomatic complaints, and is also positively associated with life satisfaction. This may be linked to the notion that religion can bring people together and can offer answers to existential questions which arise during adolescence [34,35]. Older schoolchildren also rate aspects of quality of life lower than their younger peers (15 or 16 years versus 12 or 13 years). That, too, is related to changes during puberty—for example the hormonal changes which cause young people to feel less balanced as they progress through puberty [59].

The most robust factor in explaining both aspects of life satisfaction of adolescents is social support; at home and from friends, teachers and classmates [40,41,42]. This study shows that schoolchildren with a sick parent experience less support from their family, teachers and classmates than schoolchildren without a sick parent and that this plays a role in their lower quality of life. This finding is indicative of a support effect (H4). The finding that adolescents with a sick parent report less support at home is likely to be related to the fact that the parents are heavily occupied by their own problems, meaning they are less capable of offering emotional support and engagement to their child(ren). The finding that these adolescents also experience less support from classmates might be explained by the limited time and space they experience in connecting with them [19].

### 4.1. Further Research

Concerning the informal care effect in this publication, we were able to describe the role of extra task hours in psychosomatic complaints and life satisfaction of adolescents with a chronically ill parent. Other research showed that some tasks give more pressure than others, for example giving personal care or doing administrative tasks [19,29]. Further research among young people who are involved in a variety of care situations and tasks could shed more light on the consequences of this for their quality of life. There are indications that living with a mentally ill parent is more burdensome for young people than living with a parent who has a physical illness [11,18]. The duration and severity of the parent’s illness and limitations may also be linked to young people’s well-being, which could be the focus of further research.

This study shows that informal support from the young person’s network and from their teachers can mitigate the negative impact of a care situation. However, with the data from this study, we could not investigate the need for and use of professional care apart from the support that young people receive from teachers. To gain more insight in the role that professional care plays would require further research among young people and professionals. Key questions could then be: is there sufficient attention and understanding within schools for care situations that young people face at home; does the help offered by care institutions align with the needs of schoolchildren [60]?

In this study, we have shown that specific hypotheses on the quality of life of schoolchildren with sick parents (H1–4) can be tested using a large representative sample and multivariate models. This allows for the generalisability of outcomes and insight in relevant correlations. However, to be able to shed more light on the causal relationship between having a sick parent and subsequent problems for schoolchildren, longitudinal studies are recommended. A major gap in our knowledge is namely how young people with a chronically ill parent at home fare in the long term [13], not just with regard to their quality of life, but also their school performance, choices in further education, first steps on the labour market, entering into relationships and family formation, and what role informal and professional care play in this process. Therefore, an important question for further research is: To what extent can help provided at a young age prevent schoolchildren who have a chronically ill parent from developing a lower quality of life later on?

### 4.2. Strengths and Limitations

A strength of this study is that it is a large-scale representative survey in which young people both with and without an ill parent are included. This allows for a thorough comparison of the quality of life of young people who do or do not grow up with an ill parent. Another advantage is that a wide range of explanations for differences in quality of life could be examined, e.g., care situation, living conditions and social support. Another important explanation is the informal care effect, which we were able to broadly analyse, since all young people were asked whether they perform tasks at home and, if so, how many hours they spend doing so. With a few exceptions, most previous studies only asked the group of young people will an ill loved one about tasks [7].

However, it is possible that we have not measured the (informal care) tasks precisely enough, and that we have, therefore, underestimated this effect. Time spent on household and caring tasks could perhaps be mapped more accurately if young people were to keep a daily record of their activities during a certain period (for example, a week) [61]. An alternative explanation for the relatively low informal care effect may be that young people derive pleasure from being able to contribute to the well-being of their sick parent, or that they see performing certain tasks as the natural thing to do [29].

This publication examined the role of having an ill parent as it relates to young people’s quality of life. Ideally, we would have also included more subjective indicators of how the young people experience such care situations in the analyses. However, this information was not available in the data. In order to accurately identify the ‘family effect’ described by Bobinac and colleagues [14] and ‘worrying about’ as part of the caring process [14], further research is needed which includes such indicators.

The data in this study were collected before the coronavirus outbreak occurred and changed the lives of young people. There are indications that young informal carers will have more problems due to the consequences of the COVID-19 pandemic [62,63]. This makes it necessary to invest in the prevention of problems and easily accessible mental health care, as well as to invest in supporting young people who live with an ill parent and young people who already have an increased risk of experiencing psychosomatic complaints and a lower quality of life. The long-term expectation is that an economic recession will occur as a result of the COVID-19 pandemic, and some of the young people will drop out of school. The question is whether this will also translate to a lower quality of life in the long term for young people with an ill parent. To answer this question, further research is needed.

## 5. Conclusions

The results of the present research suggest that there are gains to be made in the support at school, because pupils with a sick parent at home feel they receive relatively little support from classmates and teachers. One means of achieving these gains would be to raise awareness at school around the theme of illness and care, for example through dedicated information sessions or workshops. Schoolchildren with a sick parent (or other sick family member) at home are then more likely to feel seen and heard and may receive more support from classmates and teachers [1].

Care professionals also have a number of tools at their disposal for delivering support to these adolescents. One crucial moment is the home interview (‘*keukentafelgesprek*’). It is often assumed that, from a certain age, children will be able to perform certain tasks in the home (‘usual care’) [13]. However, the protocols and expectations of usual care may vary from municipality to municipality [64]. Thus, the specific provision of professional support will differ across townships, in the sense that in one township, it may be more extensive and accessible than in another. The question is whether the home interviews take into account the fact that there is more going on in the life of the child than simply the care situation, such as school, sport and meeting friends—all activities which are important for their development. When referring someone for help, the focus should not only be on what young people do or can do in the home, but above all on the emotional burden that growing up with a sick parent brings. Care professionals who are already working with these families ought to keep a regular eye on whether parents and other household members are able to provide sufficient support to the adolescents.

Apart from these differences, there is much that professionals have to offer. A parenting course may not appeal to everyone, but could be offered as a means of restoring some equilibrium to the domestic situation. Some of the points addressed here could include: How do you hold the family together as a parent in a family with a care situation? How do you divide your attention between the sick parent and the child and other children? And how can you give sufficient attention to the adolescent as a parent who is ill?

In this study, we looked at the quality of life of schoolchildren living with a chronically ill parent as compared to peers who are not confronted with such a situation. Leaving aside all manner of other important factors—including the extra time these children spend on tasks, that they are more likely to be female, more likely to be growing up in single parent or stepfamilies and experience less social support—having a chronically ill parent at home is found to be associated with more psychosomatic complaints and lower life satisfaction among schoolchildren. Identifying these issues at an early stage and giving these young people the support they need is important in preventing their complaints from becoming structural and problematic, and enabling them to develop as optimally as possible.

## Figures and Tables

**Table 1 ijerph-19-07043-t001:** Description of the extra hours of tasks at home, life circumstances and characteristics, support, and quality of life of schoolchildren (aged between 12 and 16) without and with a chronically sick parent—tested for significant differences in the column *p*-Value—and of the total study population (*n* = 6122).

	Schoolchildren without Sick Parent (*n* = 5470)	Schoolchildren with Sick Parent (*n* = 652)	*p*-Value	Total Group (*n* = 6122)
Extra hours of tasks at home (>4 h) (%)	12.2	18.9	<0.0005	12.9
Life circumstances and characteristics				
Female (%)	47.0	57.0	<0.0005	48.1
Age (Mean; sd)	13.9 (1.4)	14.0 (1.4)	n.s.	13.9 (1.4)
Non-native Dutch (%)	22.4	21.7	n.s.	22.3
School level			n.s.	
Lowest type secondary education (ref.) (%)	18.1	16.1	-	17.9
Low to middle type secondary education (%)	28.2	30.0	n.s.	28.4
Middle to high type secondary education (%)	24.5	26.0	n.s.	24.7
Highest type secondary education (%)	29.2	28.0	n.s.	29.0
Family affluence			<0.005	
Low (ref.) (%)	9.0	11.6	-	9.3
Middle (%)	47.7	53.1	n.s.	48.3
High (%)	43.3	35.3	<0.005	42.5
Not living with both parents in one house (%)	22.2	26.3	<0.05	22.6
Religion (very) important (%)	24.8	25.8	n.s.	24.9
Social support				
Support at home (Mean; sd)	5.5 (1.5)	5.2 (1.6)	<0.005	5.5 (1.5)
Support from friends (Mean; sd)	5.2 (1.4)	5.0 (1.5)	n.s.	5.2 (1.4)
Support from teachers (Mean; sd)	3.4 (0.9)	3.3 (0.9)	<0.01	3.3 (0.9)
Support from classmates (Mean; sd)	3.7 (0.9)	3.5 (0.9)	<0.0005	3.7 (0.9)
Quality of life				
Psychosomatic complaints (scale, Mean; sd)	3.9 (1.7)	4.5 (1.7)	<0.0005	4.0 (1.7)
Headache (Mean; sd)	3.2 (2.8)	4.0 (3.0)	<0.0005	3.3 (2.8)
Stomach ache (Mean; sd)	2.4 (2.2)	2.8 (2.5)	<0.0005	2.4 (2.2)
Backache (Mean; sd)	2.7 (2.6)	3.4 (3.0)	<0.0005	2.7 (2.6)
Feeling low (Mean; sd)	2.6 (2.5)	3.3 (2.9)	<0.0005	2.7 (2.6)
Feeling irritable or bad tempered (Mean; sd)	4.1 (2.7)	4.7 (2.8)	<0.0005	4.2 (2.7)
Feeling nervous (Mean; sd)	3.6 (2.6)	3.9 (2.7)	<0.05	3.6 (2.6)
Difficulties in getting to sleep (Mean; sd)	3.8 (3.3)	4.7 (3.7)	<0.0005	3.9 (3.3)
Feeling dizzy (Mean; sd)	2.5 (2.5)	3.0 (2.8)	<0.0005	2.5 (2.5)
Feeling tired (Mean; sd)	3.6 (3.2)	4.6 (3.5)	<0.0005	3.7 (3.2)
Feelings of exhaustion (Mean; sd)	3.6 (3.1)	4.4 (3.3)	<0.0005	3.7 (3.1)
Life satisfaction (Mean; sd)	7.7 (1.5)	7.2 (1.6)	<0.0005	7.6 (1.6)

**Table 2 ijerph-19-07043-t002:** Stepwise multilevel regression of having a chronically sick parent at home, extra task hours at home, life circumstances and characteristics, and social support on psychosomatic complaints (scale) of schoolchildren aged between 12 and 16 (unstandardised coefficients, *n* = 5932–6100).

	Model 1	Model 2	Model 3	Model 4
With a sick parent at home (ref.: without sick parent)	0.58 **	0.55 **	0.47 **	0.37 **
Extra hours of tasks at home (ref.: No extra tasks)		0.40 **	0.37 **	0.29 **
Life circumstances and characteristics				
Female (ref.: Male)			0.77 **	0.79 **
Age			0.10 **	0.04 *
Non-native Dutch (ref.: Dutch)			0.25 **	0.18 **
School level (ref.: Highest type secondary education)				
Lowest type secondary education			−0.13	−0.11
Low to middle type secondary education			−0.07	−0.07
Middle to high type secondary education			0.04	0.00
Family affluence (ref.: High)				
Low			0.07	−0.03
Middle			0.00	−0.03
Not living with both parents in one house (ref.: Living with both parents in one house)			0.24 **	0.16 **
Religion (very) important (ref.: religion not import.)			−0.19 **	−0.13 *
Support				
Support at home				−0.22 **
Support from friends				−0.04 *
Support from teachers				−0.26 **
Support from classmates				−0.13 **
Intercept	3.94 **	3.89 **	2.11 **	5.82 **
Variance at class level	0.12 **	0.11 **	0.06 **	0.04 **
Variance at individual level	2.69 **	2.68 **	2.52 **	2.27 **
Variance explained compared to empty model (%)	1.14	1.62	7.50	16.65
N	6100	6100	5940	5932
Interclass correlation class	0.04	0.04	0.02	0.02

* *p* < 0.05, ** *p* < 0.01.

**Table 3 ijerph-19-07043-t003:** Stepwise multilevel regression of having a sick parent at home, extra hours of tasks at home, life circumstances and characteristics, and social support on life satisfaction of schoolchildren aged between 12 and 16 (unstandardised coefficients, *n* = 5931–6094).

	Model 1	Model 2	Model 3	Model 4
With a sick parent at home (ref.: without a sick parent)	−0.47 **	−0.45 **	−0.37 **	−0.25 **
Extra hours of tasks at home (ref.: No extra tasks)		−0.34 **	−0.28 **	−0.18 **
Life circumstances and characteristics				
Female (ref.: Male)			−0.50 **	−0.56 **
Age			−0.18 **	−0.11 **
Non-native Dutch (ref.: Dutch)			−0.02	0.06
School level (ref.: Highest type secondary education)				
Lowest type secondary education			0.07	0.09
Low to middle type secondary education			−0.10	−0.09
Middle to high type secondary education			−0.11 *	−0.06
Family affluence (ref.: High)				
Low			−0.45 **	−0.30 **
Middle			−0.23 **	−0.17 **
Not living with both parents in one house (ref.: Living with both parents in one house)			−0.37 **	−0.27 **
Religion (very) important (ref.: Religion not import.)			0.22 **	0.14 **
Support				
Support at home				0.30 **
Support from friends				0.09 **
Support from teachers				0.17 **
Support from classmates				0.21 **
Intercept	7.67 **	7.72 **	10.63 **	6.25 **
Variance at class level	0.12 **	0.11 **	0.04 **	0.02 **
Variance at individual level	2.26 **	2.25 **	2.13 **	1.75 **
Variance explained compared to empty model (%)	0.88	1.23	6.49	23.49
N	6094	6094	5939	5931
Interclass correlation class	0.05	0.05	0.02	0.01

* *p* < 0.05, ** *p* < 0.01.

## Data Availability

The dataset analysed during the current study is not publicly available but is available from the corresponding author upon reasonable request.

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
