# Peer review of "Quality of Life of Schoolchildren Living with a Long-Term Sick Parent: The Role of Tasks at Home, Life Circumstances and Social Support"

_ijerph, 2022, doi:10.3390/ijerph19127043_

Round 1
Reviewer 1 Report
A large study of a topic previously studied by others.
Results Table 1 add actual P values to the table and then shorten the accompanyng text.
The main results relate to psychosomatic complaints. The results for each of the 10 psychosomatic complaints should be presented. I am surprised at the huge statistical difference betwen the two groups(P<0.0001) considering the large SD (1.7). The modelling is dependent upon these results, so important to confirm the difference.
I cannot comment on the statistical analysis and modelling and advise a statistical review
Author Response
Dear reviewer 1,
Thank you for your useful suggestions. p-values have been added now in Table 1 (only significant ones). In this table (concerning bivariate analyses) now also results for the 10 separate psychosomatic complaints are shown. The statistical difference between the adolescents with and without an ill parent is indeed huge for many outcome measures. The accompanying text of Table 1 was shortened. For clarity, efficiency and also for robustness, in the multivariate analyses (in Tables 2 and 3) still only the scale Psychosomatic complaints was used.
Reviewer 2 Report
This is a report of a study conducted in 2017 on Dutch adolescents between 12 and 19 comparing life satisfaction regarding a number of variables of 5,470 schoolchildren who did not have a parent with a chronic illness and 652 who did have a parent with a chronic illness. There are four hypotheses put forward by the authors regarding this study and each is supported to some degree by the results. The authors offer suggestions for further research and how young people who act as carers for parents suffering from long-term illness can be treated more equitably, permitting them better life satisfaction.
The strengths of this paper are that it is well written and very thorough in the approach it took to this investigation. One weakness is that the study was conducted in 2017 and may not reflect the current situation in Holland with respect to young people as carers. This weakness is hinted at by the authors when they suggest that life satisfaction may have further decreased in this subject group as a result of COVID-19. Another weakness is that, in making use of a number of different theories and models in designing their survey, the authors have not explained why they have chosen the selected models and theories. Why they have done so needs to be explained.
What follows are the page by page suggested edits for the submission.
Page by page suggested edits
Page 2
“Hypotheses” is not a section heading offered as part of the MDPI template. Some of what is included in this section of the authors is either part of the Introduction or the Materials and Methods. This section 2 should be broken down and redistributed between the Introduction and the Materials and Methods.
“Nevertheless, the desire is also frequently expressed to conduct more theory-driven research”— need a reference for this claim.
“According to the transactional ecological development model” —there needs to be a description of this model and in what manner the authors will make use of it for their study.
“in terms of having more psychological problems.”—need to define what is being included as psychological problems since this is used as a “narrow definition”.
Page 3
Delete “(they are the only ones who can help an ill parent, after all)”.
Given that the second hypothesis assumes that times spent on tasks at home lowers quality of life of schoolchildren, a definition of quality of life needs to be provided. The authors seem to be assuming that completing tasks at home necessarily lowers schoolchildren’s quality of life. If this is so, research needs to be presented to support this assumption. If it is not so, then it has to be made clear exactly what is meant by quality of life.
Page 4
“according to the transactional development and life course model and social resource theory”— there needs to be a description of this model and theory and in what manner the authors make use of them for their study.
Page 5
“The questionnaires were administered in a classroom-based test setting under the supervision of research assistants in the fall of 2017.”— this research was conducted 4.5 years ago. A reason should be provided as to why the results of this study have taken this long to be brought for publication.
“(‘Are you a boy or a girl?’)”—It should be made clear that there was no option provided for transgender students to select.
Page 6
“(about
-0.55), and were highly skewed on one statement (< -1.2)”—please place this information on the same line and eliminate the indent.
Page 7
“Cantril’s self-anchoring ladder was used to measure life satisfaction”—why was this chosen as the way to represent life satisfaction?
Table 1 headings—rather than have a hyphen between par-ent, place “parent” on the third line. The heading for the second column should be the same width as the first column on the left. As such, “Schoolchildren” should be all on one line. “Total” needs to be centred.
Page 8
Please use Model 1 in the text description, rather than first model, to correspond with Table 2.
Page 9
Please use Model 3 in the text description, rather than third model, to correspond with Table 2.
Page 11
“sex is once again a key predictor”—change sex to gender.
“young people become less satisfied with their lives as they grow older”—need a reference for this claim unless this was found in the data. If this is so, say so.
Page 13
As there is no section for “Further Research” in the MDPI template, please incorporate this into the discussion section.
“their
teachers can mitigate the negative impact of a care situation. However, with the data” —please place this information on the same line and eliminate the indent.
Page 14
As there is no section for “Implications” in the MDPI template, please incorporate this into the conclusions section.
Page 15
Delete the additional space before reference 2.
Page 16
Fix the formatting of reference 13.
There are a number of references missing the doi. Please include the doi for each journal article.
Page 17
Fix the formatting of references 42, 43, 48 and 57.
There are a number of references missing the doi. Please include the doi for each journal article.
Author Response
Comments to reviewer 2, manuscript IJERPH 1753069
Reviewer 2
Comments and Suggestions for Authors
This is a report of a study conducted in 2017 on Dutch adolescents between 12 and 19 comparing life satisfaction regarding a number of variables of 5,470 schoolchildren who did not have a parent with a chronic illness and 652 who did have a parent with a chronic illness. There are four hypotheses put forward by the authors regarding this study and each is supported to some degree by the results. The authors offer suggestions for further research and how young people who act as carers for parents suffering from long-term illness can be treated more equitably, permitting them better life satisfaction.
The strengths of this paper are that it is well written and very thorough in the approach it took to this investigation. One weakness is that the study was conducted in 2017 and may not reflect the current situation in Holland with respect to young people as carers. This weakness is hinted at by the authors when they suggest that life satisfaction may have further decreased in this subject group as a result of COVID-19. Another weakness is that, in making use of a number of different theories and models in designing their survey, the authors have not explained why they have chosen the selected models and theories. Why they have done so needs to be explained.
Overall answer: Thank you for your encouraging and helpful feedback. Your suggestions were well taken and we seriously considered each of them. We gratefully used them to improve our manuscript, especially as they made us realize what was still unclear. In the introduction we explained the choice of the theories, e.g., because they mostly are broad theories incorporating many aspects of young people’s lives. We will respond to each of the comments raised by reviewer 2 in detail below.
What follows are the page by page suggested edits for the submission.
Page by page suggested edits
Page 2
“Hypotheses” is not a section heading offered as part of the MDPI template. Some of what is included in this section of the authors is either part of the Introduction or the Materials and Methods. This section 2 should be broken down and redistributed between the Introduction and the Materials and Methods.
Answer: You are right; we now have positioned the four hypothesis in the introduction and renumbered the following sections.
“Nevertheless, the desire is also frequently expressed to conduct more theory-driven research”— need a reference for this claim.
Answer: Thank you for this suggestion. The desire to have more theory driven publications was formulated by Joseph and colleagues and Bobinac and colleagues, so we replaced source [13] one sentence backwards and also included source [14].
“According to the transactional ecological development model” —there needs to be a description of this model and in what manner the authors will make use of it for their study.
Answer: We highly appreciate the reviewer’s suggestion to explain the choice for the transactional ecological developmental model. It was not clear in the manuscript. We have added at page 2 a more extensive explanation: This reasoning as well as the other hypotheses are deduced from the ecological model of Bronfenbrenner [16]; a broad developmental theory describing the development of a child as a dynamic process between the child, parent(s), other significant persons (like friends and teachers), and the (broader) environment including living conditions. According to this model, a care situation at home is an unfavourable condition that creates a risk for lower quality of life of young people [15-17]. Also at pages 2 and 3 more is explained about the theoretical models
“in terms of having more psychological problems.”—need to define what is being included as psychological problems since this is used as a “narrow definition”.
Answer: Emotional and behavioural problems were added to explain the narrow definition of well-being in the study mentioned.
Page 3
Delete “(they are the only ones who can help an ill parent, after all)”.
Answer: Done
Given that the second hypothesis assumes that times spent on tasks at home lowers quality of life of schoolchildren, a definition of quality of life needs to be provided. The authors seem to be assuming that completing tasks at home necessarily lowers schoolchildren’s quality of life. If this is so, research needs to be presented to support this assumption. If it is not so, then it has to be made clear exactly what is meant by quality of life.
Answer: Thank you for pointing this out. We added three references. At the top of page 2 we included some text, more clearly defining both aspects of quality of life.
Page 4
“according to the transactional development and life course model and social resource theory”— there needs to be a description of this model and theory and in what manner the authors make use of them for their study.
Answer: We indeed did not elaborate on this theoretical input. In the new version of the manuscript we explain which aspects of the named models are used: In the transactional development model and the life course model the focus lies on completing successive development tasks (e.g. developing one's own identity and obtaining a basic qualification) and the role of support from parents and significant others in this regard. The social resource model gives information on how the structures of provision effects the positive or negative outcomes of informal care giving. The present study focuses on support within the family from parents, brothers and sisters, as well as support from friends, classmates and teachers
Page 5
“The questionnaires were administered in a classroom-based test setting under the supervision of research assistants in the fall of 2017.”— this research was conducted 4.5 years ago. A reason should be provided as to why the results of this study have taken this long to be brought for publication.
Answer: In 2019 we have written a report on youngsters with an ill family member for our own organization (The SCP) and for policy makers, published in 2020. As our main output is not to publish in (international) scientific journals, at that time we did not prepare a scientific publication. After we received in 2021 a request from IJERPH (professor Pakenham and Landi) to participate in the special Issue "Effects of Parental Physical and Mental Illness on Children and Adolescents" we started this process. However, in order to match the central research question of the special issue, we had to do additional analyses. So it took a while to get this manuscript finished.
“(‘Are you a boy or a girl?’)”—It should be made clear that there was no option provided for transgender students to select.
Answer: We added this suggestion in the text.
Page 6
“(about
-0.55), and were highly skewed on one statement (< -1.2)”—please place this information on the same line and eliminate the indent.
Answer: Done
Page 7
“Cantril’s self-anchoring ladder was used to measure life satisfaction”—why was this chosen as the way to represent life satisfaction?
Answer: We have done secondary analyses on HBSC data, which included this well-known measure. In the text we added: a well-known measure and repeatedly used in the HBSC survey [48, 53, 55].
Table 1 headings—rather than have a hyphen between par-ent, place “parent” on the third line. The heading for the second column should be the same width as the first column on the left. As such, “Schoolchildren” should be all on one line. “Total” needs to be centred.
Answer: this was done
Page 8
Please use Model 1 in the text description, rather than first model, to correspond with Table 2.
Answer: this was done
Page 9
Please use Model 3 in the text description, rather than third model, to correspond with Table 2.
Answer: this was done
Page 11
“sex is once again a key predictor”—change sex to gender.
Answer: Done
“young people become less satisfied with their lives as they grow older”—need a reference for this claim unless this was found in the data. If this is so, say so.
Answer: It was found in the data, and now said in the description of the outcomes, also we included reference 48 (also about age differences in the international HBSC-study)
Page 13
As there is no section for “Further Research” in the MDPI template, please incorporate this into the discussion section.
Answer: done
“their
teachers can mitigate the negative impact of a care situation. However, with the data” —please place this information on the same line and eliminate the indent.
Answer: Done
Page 14
As there is no section for “Implications” in the MDPI template, please incorporate this into the conclusions section.
Answer: Done
Page 15
Delete the additional space before reference 2.
Answer: Done
Page 16
Fix the formatting of reference 13.
There are a number of references missing the doi. Please include the doi for each journal article.
Answer: the formatting of reference 13 was fixed and we added the doi as often as possible
Page 17
Fix the formatting of references 42, 43, 48 and 57.
Answer: we tried to fix the formatting of these references
There are a number of references missing the doi. Please include the doi for each journal
Answer: The doi has been added as often as possible